# Effects of a Forest Therapy Program on Physical Health, Mental Health, and Health Behaviors

**Hae-ryoung Chun** [1] , **Inhyung Cho** [1] , **Yoon Young Choi** [1] , **Sujin Park** [2] , **Geonwoo Kim** [2] and **Sung-il Cho** [1,*]

1 Graduate School of Public Health, Seoul National University, 1 Gwanak-ro, Gwanak-gu, Seoul 08826, Republic of Korea; mamimihae@gmail.com (H.-r.C.); ihcho04@snu.ac.kr (I.C.); young.feb17@snu.ac.kr (Y.Y.C.)
2 National Institute of Forest Science (NIFoS), 57, Hoegi-ro, Dongdaemun-gu, Seoul 02455, Republic of Korea; snowshoe@korea.kr (S.P.); bkim5020@korea.kr (G.K.)
* Correspondence: persontime@hotmail.com

**Abstract:** (1) Background: Although interest in the health-promoting effects of forest therapy is increasing, few researchers have investigated the mid-long-term impact of such therapy on health indicators or exercise behaviors. We explored changes in physical health, mental health, and exercise behaviors 1, 2, and 4 weeks after a forest therapy program concluded. We sought to establish a solid foundation for such programs and a standardized evaluation system. (2) Method: We measured the blood pressure and heart rate variability of 99 adults before and after participation in a forest therapy program. We used the State-Trait Anxiety Inventory to assess anxiety, the Beck Depression Inventory to evaluate both anxiety and depression, the Profile of Mood States to explore mood, the Euro-Quality of Life-5 Dimension scale to assess the overall quality of life, and the Positive and Negative Effect Schedule to measure positive and negative mood. We employed the Global Physical Activity Questionnaire to determine exercise time, intensity, and changes in exercise type before the program and 1–4 weeks after program completion. (3) Results: Anxiety, depression, mood, quality of life, heart rate, and blood pressure control improved significantly after the program. The reduced depression and increased medium-intensity exercise time persisted for 1, 2, and 4 weeks after the end of the program. (4) Conclusions: We tracked various health indicators and clearly distinguished those that were useful in the short term from those more appropriate for evaluation in the long term. This is the first report to show that a forest therapy program affects exercise behavior; this suggests that health behaviors should be continuously tracked.

**Keywords:** Shinrin-yoku; forest healing; physiological; psychological; health behavior

## 1. Introduction

Forest therapy is a nature-centered intervention that utilizes various natural elements of forest environments, including landscape, phytoncides, anions, sounds, sunlight, oxygen levels, plants, water, nutrition, psychological therapy, and climate [1]. Forest therapy programs that utilize these elements promote both mental and physical health. According to review papers related to forest therapy programs, the most prominent therapeutic effects include a decrease in blood pressure, alleviation of depressive symptoms [2], and improvement in mood assessed by POMS [3]. Interest in forest therapy programs is increasing. In the case of South Korea, the number of national forests in which people can participate in forest therapy programs has expanded. The number of visitors was only 1067 in 2009, which increased to 1.7 million in 2015 and 1.8 million in 2019; even in 2020 (the year of a global pandemic), there were 1.5 million visitors [2].

However, the long-term benefits of forest therapy remain unclear [3,4]. One study evaluated health outcomes immediately after program completion and 3, 4, and 7 days later [5], while another explored the effects immediately after program completion and

2 and 4 weeks later [6]. In contrast to most forest therapy program studies that have predominantly examined short-term effects, a few preceding studies have investigated long-term effects up to 4 weeks after program completion [6,7]. Specifically, Park (2021) considered the long-term effects over four weeks to overcome the limitation of primarily focusing on the short-term effects of forest therapy programs [7]. However, the long-term effects have received little attention. One qualitative study interviewed participants to explore long-term health behavior changes after forest therapy [8]. Yi et al., in a systematic review and meta-analysis, found that only 3 of 17 studies examined the effects of forest therapy after one week [9]. Likewise, quantitative data supporting forest therapy's long-term impact are scarce.

Another aspect of forest therapy is that experiencing forest therapy not only enhances mental health but also could lead to a change in health behavior. Some authors measured general health [10] and the quality of life [11] after such therapy but did not connect with changes in physical activity. Increased physical activity may mediate the long-term effects of forest therapy on physical and mental health, but relevant research is lacking. Physical activity promotes physical health, encourages healthier lifestyles, and improves mental health. Daily walking improved health, self-awareness of body composition, and exercise motivation [12]. Another study only reported increased physical activity after forest therapy [13]. Thus, a comprehensive investigation of the possible long-term enhancement of physical activity and physical and mental health is required.

Very few studies obtained consent from participants who had already registered for a forest therapy program, with most research recruiting participants based on experimental designs and conducting studies for specific periods. Most research on the effects of forest therapy has been conducted with more specific participants; consequently, participant recruitment must be undertaken more stringently. To assure external validity and practical applicability, however, real-world evaluations must be conducted to assess various participants [14,15].

The health-promoting effects of forest therapy have already been established, but the long-term effects of forest therapy using standardized indicators have not been evaluated. Therefore, this study aims to identify health indicators that demonstrate long-term effects. We evaluated the effects of forest therapy on health at 1, 2, and 4 weeks after the completion of the program, focusing on changes in exercise behavior. Given that previous research has demonstrated improvements in depression alleviation and quality of life up to 4 weeks after program completion [6], we anticipated similar positive effects in our study.

## 2. Materials and Methods

### 2.1. Pilot Study

A total of 99 self-recruited participants stayed at the National Yeongju Forest Therapy Center for two days and one night, experiencing at least one indoor and two outdoor programs. The outdoor programs involved a 2-h guided forest walk led by a forest therapy guide, during which participants listened to bird songs, smelled the scent of trees, and touched trees while enjoying the scenery. Indoor programs included simple exercises using basic equipment and meditation. We conducted HRV, blood pressure measurements, and surveys upon entry to the Yeongju Forest Therapy Center for the forest therapy program. We conducted HRV (Heart Rate Variability), blood pressure measurements, and surveys just before leaving the Forest Therapy Center after completing all programs. We measured blood pressure using the EASY X800 digital machine (SELVAS Healthcare, Inc., Seoul, Republic of Korea) and heart rate variability using the SA6000 machine (MEDICORE Co., Hanam-si, Republic of Korea). Both devices are securely fixed to a table. The heart rate variability measurement device automatically conducts a 3-min measurement by attaching electrodes to the left and right wrist areas and the left ankle area. A follow-up online survey (questionnaires only) was performed 1, 2, and 4 weeks after return to daily life; there were 46, 32, and 17 responses, respectively (Figure 1). The following data collection process was carried out.

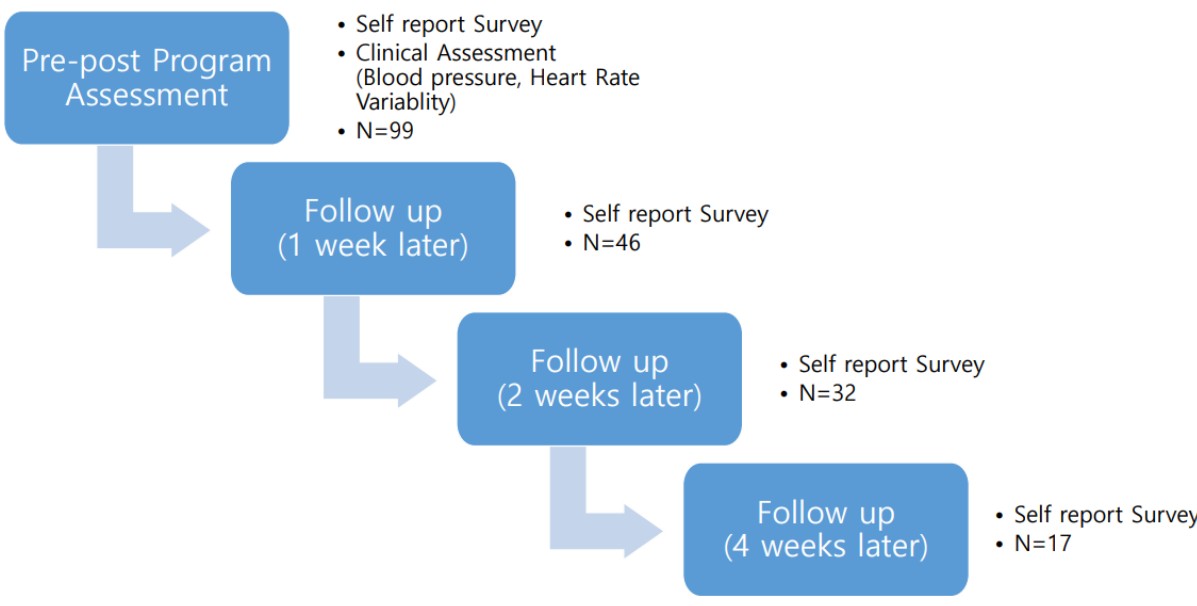

**Figure 1.** Research flow.

- After participants arrived and registered, they received an explanation of the research procedure and gave written informed consent to provide physical health data. The physiological data are not collected continuously but only before and after the program.
- Before the assessment, groups with five or more participants were divided into smaller groups of five participants each. This was partly due to a shortage of measuring devices and to prevent sharing individual results.
- Measurements were taken before and after the program. All questionnaires were self-administered; the heart rate and blood pressure were measured using an HRV (SA6000) device and an automatic EASY X800 blood pressure monitor after each participant had been sitting in a comfortable chair for at least 15 min. HRV was measured only once, while blood pressure was measured twice after the readings stabilized and recorded along with participant names (or other identifications). The average of the two blood pressure measurements was used in the analysis.
- After the program, all measurements were repeated identically.
- The data were collected and organized.
- Personal information was anonymized.
- Online email surveys were also sent to the initial participants 1, 2, and 4 weeks after program completion.

Participants

A total of 99 participants in seven groups participated (one individual participant, two groups from the Gyeongbuk Social Welfare Association, one team from the Eunpyeongol Child Protection Association, a Kyobo Education Foundation Family group, a group of firefighters, and a group from Baejae University).

*2.2. Methodology for Health Measurement Tools Selection*

We systematically reviewed the literature measuring physical and mental health and built a research protocol [3]. We selected five mental health evaluation tools that met the following criteria: few questions, completion in <10 min, published in Korean, and appropriate for administration before or after 1-day or 1-night stays. To measure physical health, we employed the equipment of the Forest Therapy Center. Six mental health, healthcare, and forestry experts were consulted when planning the research.

### 2.3. Health Measurement Tools

We recorded participant telephone numbers, email addresses, countries and regions of origin, gender, and age. Heart rate was measured using an HRV device of the Forest Therapy Center, and a blood pressure monitor measured blood pressure and pulse rate. In the HRV device, the x-axis of the autonomic nervous system stability represents LF, while the y-axis represents HF. Stress resistance is indicated by SDNN, stress index by PSI, fatigue by LF, mean heart rate by mean HR, and heart stability by HF. The State-Trait Anxiety Inventory (STAI-X) from the publisher (www.mindgarden.com, accessed on 15 August 2020) was used to evaluate anxiety, the Beck Depression Inventory (BDI) by Harcourt Assessment Incorporated (Pearson Education plc) to assess depression, the Profile of Mood States-Brief (POMS-B) by Multi-Health Systems Inc. (MHS) to measure mood, the Euro-Quality of Life-5 Dimension (EQ-5D) scale by EuroQol Office to explore quality of life, and the Positive Affect and Negative Affect Schedule (PANAS) by the American Psychological Association to evaluate positive and negative mood states. For the STAI-X, scores < 52 indicate a low level of anxiety, scores between 52 and 56 indicate slightly elevated anxiety, scores between 57 and 61 a high level of anxiety, and scores of 62 and above denote a very high level of anxiety. The BDI score ranges from 0 to 63, where scores < 9 indicate a non-depressed state, scores between 10 and 15 reflect mild depression, scores between 16 and 23 indicate moderate depression, and scores between 24 and 63 denote severe depression. The POMS-B sub-indices include Tension-Anxiety (TA), Depression (D), Anger (A), Vigor (V), Confusion (C), and Fatigue (F); the Total Mood Disturbance score is given by the sum of the TA, D, AH, F and C scores minus that of V. The maximum POMS-B score is 25; higher scores indicate poorer mood. The maximum EQ-5D score is 1; higher scores indicate better quality of life. The PANAS uses subindices to evaluate positive and negative moods. We employed the Global Physical Activity Questionnaire (GPAQ) to analyze physical activity [16], i.e., high- and moderate-intensity exercises and the daily and weekly frequencies. High-intensity exercise increases respiration and the heart rate more than moderate-intensity activity. High-intensity exercise in GPAQ includes running, aerobic exercise, and cycling at a fast pace. In contrast, moderate-intensity exercise comprises activities like cycling at a regular pace and playing doubles tennis.

### 2.4. Data Extraction and Analysis

The online surveys were submitted via Google Forms. Blood pressure and heart rate values indicated on the displays were checked immediately after measurement. Mental health (anxiety assessed by the STAI-X, depression by the BDI, mood by the POMS-B, quality of life by the EQ-5D, and static and dynamic emotions by the PANAS) was assessed before and after the program and 1, 2, and 4 weeks after program completion. The GPAQ was administered before the program and 1, 2, and 4 weeks after completion. Blood pressure and HRV were measured before and after the program. Table 1 presents the InBody measurement results as mean values for each group. In the remaining tables, we conducted the Wilcoxon signed-rank test to compare the differences between the physical mental health variables and physical health behavior before and after the program. Table 2 compares the health status before and immediately after the program, Table 3 compares the health status before the program and one week after program completion, Table 4 compares the health status before the program and two weeks after program completion, and Table 5 examines the changes in health status before the program and four weeks after program completion. All analyses were conducted using R (version 4.0.3) and R Studio software version 4.3.0.

**Table 1.** Results of demographic analysis by group.

| Group * | Total | A | B | C | D | E | F | G |
|---|---|---|---|---|---|---|---|---|
| Male | | | | | | | | |
| *N* | 27 | 6 | 4 | 1 | 0 | 2 | 2 | 12 |
| Age (years) | 29.6 | 40.0 | 25.0 | 30.0 | - | 40.0 | 40.0 | 22.5 |
| Height (cm) | 173.2 | 171.5 | 175.8 | 175.0 | - | 175.3 | 170.5 | 175.8 |
| Weight (kg) | 81.0 | 74.6 | 82.7 | 130.7 | - | 71.9 | 81.4 | 66.7 |
| BMI (kg/m$^2$) | 26.8 | 25.0 | 26.3 | 42.7 | - | 23.3 | 28.0 | 22.2 |
| Female | | | | | | | | |
| *N* | 72 | 18 | 16 | 14 | 2 | 2 | 8 | 12 |
| Age (years) | 35.8 | 40.6 | 31.3 | 41.4 | 55.0 | 40.0 | 42.5 | 20.0 |
| Height (cm) | 160.8 | 159.5 | 159.5 | 163.1 | 160.0 | 166.8 | 161.2 | 161.8 |
| Weight (kg) | 61.8 | 57.2 | 66.1 | 64.1 | 63.4 | 60.6 | 59.7 | 55.2 |
| BMI (kg/m$^2$) | 24.4 | 22.4 | 26.0 | 24.1 | 24.7 | 21.8 | 26.7 | 21.3 |

* Group A represents individuals who applied and came in groups of less than 2. Group B is the Gyeongbuk Social Welfare Staff Cooperative-1 group, and Group C is the Gyeongbuk Social Welfare Staff Cooperative-2 group. D represents the Korean Association for Child Welfare group, E is the Kyobo Education Foundation Family Group, F represents the fire department group, and G represents university students.

**Table 2.** Changes in the health status of 99 participants before and after the program.

| Tool [†] | Before | | After | | Wilcoxon |
|---|---|---|---|---|---|
| | Mean | (SD) [‡] | Mean | (SD) [‡] | *p*-Value |
| STAI | 40.45 | (10.90) | 32.74 | (9.19) | <0.01 |
| BDI | 9.30 | (7.95) | 5.30 | (6.28) | 0.00 |
| POMS | 13.91 | (22.39) | 2.96 | (16.47) | 0.00 |
| T-A | 4.54 | (4.83) | 2.33 | (3.22) | 0.00 |
| D | 3.43 | (4.35) | 1.79 | (2.97) | 0.00 |
| A-H | 3.43 | (4.42) | 1.54 | (3.02) | 0.00 |
| C | 5.00 | (3.48) | 3.86 | (2.76) | 0.00 |
| F | 5.10 | (4.55) | 2.94 | (3.37) | 0.00 |
| V | 7.58 | (5.18) | 9.49 | (5.22) | 0.00 |
| EQ5D | 0.90 | (0.12) | 0.94 | (0.10) | 0.00 |
| PanasP | 22.64 | (7.52) | 24.16 | (7.57) | 0.01 |
| PanasN | 17.44 | (7.80) | 15.10 | (6.71) | 0.00 |
| SBP | 123.53 | (17.69) | 119.59 | (16.79) | 0.00 |
| DBP | 75.69 | (13.21) | 72.89 | (11.96) | 0.00 |
| HRV * | | | | | |
| Activity | 94.67 | (24.84) | 98.58 | (19.87) | 0.27 |
| Balance | 67.40 | (44.42) | 59.04 | (45.80) | 0.42 |
| Resist | 95.59 | (22.90) | 98.06 | (18.02) | 0.45 |
| Stress | 98.28 | (20.54) | 96.73 | (22.16) | 0.18 |
| Fatigue | 103.50 | (22.52) | 104.15 | (20.67) | 0.75 |
| Heart Rate | 81.18 | (17.05) | 74.98 | (12.70) | 0.00 |
| Stability | 106.58 | (20.79) | 106.86 | (21.25) | 0.66 |

* $p < 0.05$; all *p*-values were calculated by comparing them with the value before; [†] METs: Metabolic Equivalents; STAI evaluates anxiety, BDI evaluates depression, and POMS evaluates mood state. The sub-indices of POMS are T-A (tension-anxiety), D (depression), A-H (anger-hostility), V (vigor), F (fatigue), and C (confusion); POMS = TA + D + AH + F + C-V. PANAS evaluates positive affect scores and negative affect scores. PANAS-P represents positive affect, while PANAS-N represents negative affect. The sub-indices of HRV are as follows: Autonomic nervous system activity, Autonomic nervous system balance, Stress resistance, Stress index, Fatigue level, Mean heart rate, Cardiac stability; [‡] StandardC-V Deviation.

**Table 3.** Changes in the health status of 46 respondents before, after, and one week after the program.

| Tool [†] | Before | After | *p* | After One Week | *p* |
|---|---|---|---|---|---|
| STAI | 41.30 | 31.61 | 0.00 * | 35.70 | 0.00 * |
| BDI | 9.46 | 5.85 | 0.00 * | 5.19 | 0.00 * |
| POMS | 14.35 | 2.11 | 0.00 * | 4.89 | 0.00 * |

**Table 3.** *Cont.*

| Tool [†] | Before | After | *p* | After One Week | *p* |
|---|---|---|---|---|---|
| T-A | 4.76 | 2.13 | 0.00 * | 2.59 | 0.00 * |
| D | 3.09 | 1.61 | 0.00 * | 1.78 | 0.00 * |
| A-H | 3.26 | 1.20 | 0.00 * | 2.11 | 0.00 * |
| C | 5.43 | 3.80 | 0.00 * | 4.30 | 0.01 * |
| F | 5.35 | 2.80 | 0.00 * | 3.07 | 0.00 * |
| V | 7.54 | 9.43 | 0.00 * | 8.96 | 0.00 * |
| EQ5D | 0.92 | 0.95 | 0.00 * | 0.96 | 0.00 * |
| PanasP | 22.11 | 24.41 | 0.01 * | 23.37 | 0.11 |
| PanasN | 17.65 | 14.61 | 0.00 * | 13.96 | 0.00 * |
| GPAQ | | | | | |
| Vigorous-intensity activity | | | | | |
| METs | 780.44 | | | 812.52 | 0.39 |
| Days | 1.82 | | | 1.26 | 0.52 |
| Hours | 32.39 | | | 42.00 | 0.29 |
| Moderate-intensity activity | | | | | |
| METs | 444.00 | | | 672.17 | 0.06 |
| Days | 2.07 | | | 2.39 | 0.09 |
| Hours | 40.78 | | | 56.20 | 0.05 * |

\* $p < 0.05$; all *p*-values were calculated by comparing them with the value before; [†] METs: Metabolic Equivalents; STAI evaluates anxiety, BDI evaluates depression, and POMS evaluates mood state. The sub-indices of POMS are T-A (tension-anxiety), D (depression), A-H (anger-hostility), V (vigor), F (fatigue), and C (confusion); POMS = TA + D + AH + F + C-V. PANAS evaluates positive affect score and negative affect score.

**Table 4.** Changes in the health status of 32 survey respondents before, after, one week after, and two weeks after the program.

| Tool [†] | Before | After | | After 1 Week | | After 2 Weeks | |
|---|---|---|---|---|---|---|---|
| | Mean | Mean | *p* | Mean | *p* | Mean | *p* |
| STAI | 40.03 | 29.09 | 0.00 * | 37.95 | 0.03 * | 36.22 | 0.06 |
| BDI | 9.12 | 5.42 | 0.00 * | 6.45 | 0.00 * | 4.8 | 0.00 * |
| POMS | 14.34 | 0.78 | 0.00 * | 9.27 | 0.01 * | 7.03 | 0.06 |
| T-A | 4.88 | 1.91 | 0.00 * | 3.73 | 0.06 | 2.91 | 0.03 * |
| D | 3.28 | 1.53 | 0.00 * | 2.68 | 0.28 | 2.44 | 0.15 |
| A-H | 3.53 | 1.22 | 0.00 * | 3 | 0.07 | 2.5 | 0.16 |
| C | 5.59 | 4.22 | 0.01* | 5.41 | 0.23 | 4.88 | 0.12 |
| F | 5.81 | 3 | 0.00 * | 4 | 0.00 * | 4.31 | 0.02 * |
| V | 8.75 | 11.09 | 0.00 * | 9.55 | 0.04 * | 10 | 0.09 |
| EQ5D | 0.92 | 0.96 | 0.00 * | 0.95 | 0.05 * | 0.93 | 0.33 |
| PanasP | 25.06 | 26.66 | 0.09 | 24.91 | 0.35 | 24.63 | 0.72 |
| PanasN | 18.19 | 14.31 | 0.00 * | 15.59 | 0.01 * | 15.44 | 0.04 * |
| GPAQ | | | | | | | |
| Vigorous-intensity activity | | | | | | | |
| METs | 943.45 | | | 731.64 | 0.7 | 556.25 | 0.88 |
| Days | 2.45 | | | 1.5 | 0.45 | 1.31 | 0.84 |
| Hours | 34.67 | | | 41 | 0.68 | 32.34 | 0.5 |
| Moderate-intensity activity | | | | | | | |
| METs | 412.67 | | | 456.36 | 0.36 | 530 | 0.1 |
| Days | 1.9 | | | 2 | 0.56 | 2.16 | 0.24 |
| Hours | 44.17 | | | 48.86 | 0.42 | 52.66 | 0.07 |

\* $p < 0.05$; all *p*-values were calculated by comparing them with the value before; [†] METs: Metabolic Equivalents; STAI evaluates anxiety, BDI evaluates depression, and POMS evaluates mood state. The sub-indices of POMS are T-A (tension-anxiety), D (depression), A-H (anger-hostility), V (vigor), F (fatigue), and C (confusion); POMS = TA + D + AH + F + C-V. PANAS evaluates positive affect score and negative affect score.

**Table 5.** Changes in the health status of 17 survey respondents before, after, one week after, two weeks, and four weeks after the program.

| Tool [†] | Before and After the Program | | | After 1 Week | | After 2 Weeks | | After 4 Weeks | |
|---|---|---|---|---|---|---|---|---|---|
| | Before | After | *p* | After | *p* | After | *p* | After | *p* |
| STAI | 40.31 | 31.06 | 0.00 * | 36.7 | 0.08 | 35.53 | 0.07 | 34.12 | 0.05 |
| BDI | 8.68 | 5.61 | 0.00 * | 5.38 | 0.01 * | 3.38 | 0.01 * | 3.76 | 0.01 * |
| POMS | 12.41 | 2.71 | 0.00 * | 5.4 | 0.06 | 5 | 0.2 | 3.53 | 0.05 |
| T-A | 4.47 | 1.88 | 0.00 * | 2.9 | 0.13 | 2.73 | 0.22 | 2.47 | 0.12 |
| D | 2.59 | 1.35 | 0.01 * | 1.7 | 0.3 | 1.47 | 0.14 | 1.24 | 0.02 * |
| A-H | 3.29 | 1.59 | 0.03 * | 1.7 | 0.03 * | 2 | 0.22 | 2 | 0.08 |
| C | 4.53 | 3.76 | 0.19 | 4.2 | 0.6 | 4.4 | 0.42 | 3.94 | 0.32 |
| F | 5.65 | 3.53 | 0.00 * | 3 | 0.02 * | 4.13 | 0.1 | 3.35 | 0.06 |
| V | 8.12 | 9.41 | 0.09 | 8.1 | 0.29 | 9.73 | 0.14 | 9.47 | 0.12 |
| EQ5D | 0.93 | 0.96 | 0.04 * | 0.96 | 0.14 | 0.92 | 0.58 | 0.96 | 0.06 |
| PanasP | 23.41 | 25 | 0.2 | 22.8 | 0.32 | 23.33 | 0.64 | 23.47 | 0.54 |
| PanasN | 16.65 | 14.12 | 0.07 | 13.5 | 0.06 | 14.73 | 0.17 | 13.71 | 0.04 * |
| GPAQ | | | | | | | | | |
| | Vigorous-intensity activity | | | | | | | | |
| METs | 464 | - | - | 769.6 | 0.43 | 600 | 0.15 | 809.41 | 0.17 |
| Days | 1.33 | - | - | 1.6 | 0.36 | 1.33 | 0.57 | 1.65 | 0.12 |
| Hours | 22.67 | - | - | 54.2 | 0.5 | 33 | 0.05 | 32.94 | 0.17 |
| | Moderate-intensity activity | | | | | | | | |
| METs | 428.75 | - | - | 544 | 0.08 | 661.33 | 0.1 | 720 | 0.01 * |
| Days | 2.38 | - | - | 2.3 | 0.67 | 2.53 | 0.42 | 2.47 | 0.31 |
| Hours | 40.94 | - | - | 63.5 | 0.05 * | 57.67 | 0.05 * | 62.06 | 0.02 * |

\* $p < 0.05$; all *p*-values were calculated by comparing them with the value before; [†] METs: Metabolic Equivalents; STAI evaluates anxiety, BDI evaluates depression, and POMS evaluates mood state. The sub-indices of POMS are T-A (tension-anxiety), D (depression), A-H (anger-hostility), V (vigor), F (fatigue), and C (confusion); POMS = TA + D + AH + F + C-V. PANAS evaluates positive affect scores and negative affect scores.

## 3. Results

Table 1 lists the demographic variables by gender; there were 27 male and 72 female participants with a median age of 29.6 and 35.8 years, respectively. The average male height, weight, and body mass index (BMI) were 173.2 cm, 81 kg, and 26.8 kg/m$^2$ respectively. The normal Korean BMI ranges from 18.5 to 23 kg/m$^2$; therefore, the males in this study were overweight. However, of the 12 group G subjects, the mean BMI of 22.2 kg/m$^2$ was within the normal range. For females, the average height was 160.8 cm, the average weight was 61.8 kg, and the average BMI was 24.4 kg/m$^2$; the females were thus slightly overweight. The InBody analysis results indicate that most participants fall within the normal range or are slightly overweight. Still, they are considered healthy and able to participate in the forest therapy program without difficulty.

Table 2 lists the mental and physical health data of all 99 participants before and after the program. In Table 2, all aspects of mental health showed significant improvements; blood pressure decreased, and heart rate also significantly decreased. In terms of mental health, anxiety and depression decreased after the program compared to before, negative mood measured by POMS decreased, quality of life increased, positive mood measured by PANAS increased, and negative mood decreased. The STAI-X showed that anxiety scores decreased after the program compared to before, as indicated by lower scores ($p < 0.01$).

According to the BDI results, depression scores decreased after the program, indicating reduced depression ($p = 0.00$). POMS, which measures negative moods, showed a decrease in negative moods after the program ($p = 0.00$). EQ5D indicated improved quality of life after program completion ($p = 0.00$), while PANAS-P, which measures positive moods, showed increased positive feelings ($p = 0.00$). Finally, PANAS-N, which measures negative feelings, showed decreased negative feelings ($p = 0.00$). The systolic blood pressure (SBP) ($p = 0.00$) and diastolic blood pressure ($p = 0.00$) decreased significantly. Of the HRV measures, activity increased compared to before the program ($p = 0.27$), stress balance decreased ($p = 0.42$), indicating poor balance, and stress resistance increased ($p = 0.45$). However, none of these indicators showed statistically significant changes. Furthermore, stress index decreased ($p = 0.18$), but fatigue increased ($p = 0.75$), and stability increased ($p = 0.66$). Nevertheless, these indicators also did not show statistically significant changes. Of the HRV measures, only heart rate decreased significantly ($p = 0.00$). In the HRV sub-indicator analysis in Supplementary Table S2, only heart rate showed a significant decrease.

The normal average baseline SBP was <120 mmHg in all subjects and decreased significantly after the program. HRV, heart stability, fatigue, stress index, stress resistance, and autonomic nervous system stability were initially normal. Autonomic nervous system activity is considered moderate when the score exceeds 90 and good when the score is >110. All activities were expected before the program. The autonomic nervous system balance should be <50. All participants were unbalanced before the program, but Group A became balanced after the program ended. Stress resistance of 90 or higher is considered average, and 110 or higher is considered good; all participants were balanced before the program. The stress index should be <110 (normal) or <90 (good); all subjects were normal before the program. The fatigue level should be ≤110 (normal) or ≤90 (good); all subjects were normal before the program. The normal heart rate is 70–83 bpm, and most groups were within the normal range before and after the program. Heart stability is normal in the range of 90–110 bpm; all subjects were normal before and after the program. However, the average heart rate decreased significantly in most groups.

Table 3 presents the results for the 46 participants who responded up to 1 week after the program ended. In terms of mental health, compared to pre-program levels, there were reductions in depression ($p = 0.00$) and anxiety ($p = 0.00$) immediately after the program. Negative mood measured by POMS decreased ($p = 0.00$), and there was an increase in quality of life ($p = 0.00$) and positive mood measured by PANAS ($p = 0.00$), with a decrease in negative mood. However, one week after program completion, only the increase in positive mood measured by PANAS did not show significant results ($p = 0.11$); other mental health improvements exhibited significant changes. According to the results from the GPAQ, there was a significant increase in moderate-intensity exercise time one week after program completion (increasing from 40.78 min to 56.20 min, $p = 0.049$). In terms of exercise, 600 metabolic equivalents (METs)/week is recommended; before the program, the average was <600 METs, but after one week, it exceeded 600 METs.

Table 4 examined the changes in the health status of 32 participants who responded up to 2 weeks after the program. Immediately after program completion, only positive mood assessed by PANAS did not show significant results, while other aspects of mental health showed improvement. One week after program completion, anxiety and depression ($p = 0.00$) decreased, negative mood assessed by POMS decreased, quality of life improved, and a reduction in negative mood assessed by PANAS was observed ($p = 0.04$). Two weeks after program completion, only significant decreases in depression and negative mood assessed by PANAS w observed. According to the results from GPAQ, no significant changes were noted.

Table 5 presents the results of the mental health and GPAQ scores of 17 participants who responded up to 4 weeks after program completion. Immediately after the program, improvements were observed in all aspects of mental health except for mood changes through PANAS (from 16.65 to 13.71, $p = 0.04$). However, from 1 week after program completion to 4 weeks after, the only significant effect that persisted was the reduction in

depression (from 8.68 to 3.76, $p = 0.01$). Furthermore, there was a significant increase in moderate-intensity activity time 1 week, 2 weeks, and 4 weeks after program completion (from 40.94 to 62.06, $p = 0.02$). Supplementary Table S1 lists the mental and physical health changes before and immediately after the program. Significant differences in the STAX, BDI, POMS, and SBP scores/values were apparent in all groups. When comparing the analysis results of each group with the analysis results in Supplementary Table S3 that did not distinguish between groups, it was found that all mental health improvements, except for mood changes assessed through PANAS and changes in quality of life, showed significant changes uniformly. The male group did not exhibit significant mood changes regarding quality of life or positive and negative moods assessed through PANAS.

In contrast, all aspects of mental health improvement were significant in females. While in the 20s and 30s age groups, all aspects of mental health improvement were observed, from the 40s to the 60s age group, significant positive mood enhancement through PANAS was not observed. Furthermore, among the groups categorized by BMI, the group classified as obese showed a significant increase only in positive emotions assessed through PANAS, while the non-obese group exhibited improvement in all aspects of mental health. According to HRV analysis results, except for the male group, all females across all age groups, regardless of obesity status, demonstrated a significant reduction in heart rate. In summary, it can be confirmed that the positive emotional enhancement effect assessed through PANAS is not significant for males, older individuals, or those who are obese.

Additionally, regarding BMI categories, the group classified as obese showed significant improvement only in positive mood assessed through PANAS. In contrast, the non-obese group improved all aspects of mental health. According to HRV analysis results, except for the male group, all females across all age groups, regardless of whether they were in the obese group, exhibited a significant reduction in heart rate. In conclusion, it can be confirmed that the positive mood enhancement assessed through PANAS is not significant for males, older individuals, or those who are obese.

## 4. Discussion

We investigated the long-term effects of forest therapy on adult mental health, physical health, and exercise behaviors. A significant decrease in depression (compared to the initial level) persisted for up to 4 weeks after the program, and the moderate-intensity exercise time and METs increased significantly for up to 4 weeks. Immediately after the program, anxiety, depression, quality of life, mood, and negative emotions decreased, as did the heart rate and blood pressure; forest therapy thus improved mental and physical health.

Earlier reports indicated that forest therapy programs reduced depression for at least four weeks; thus, forest therapy or simply being in the forest lowers anxiety and depression even after subjects exit the forest [6]. One report found that repeated forest therapies reduced anxiety levels [17], but no study has shown that a single forest therapy session reduces anxiety in the long term. However, this therapy reduced stress and depression and improved quality of life for up to 4 weeks after the program ended [6]. Other studies found that the improvements in depression and stress persisted for up to 4 weeks and that of quality of life for up to 2 weeks [6]. While no studies have evaluated the effects of forest therapy programs on depression and quality of life in healthy individuals over a 4-week period, some previous research has examined postmenopausal women. The results indicate that in the previous study, the quality of life showed a significant positive change up to 2 weeks later. However, in this study, since the quality of life was already high before the program, there was no significant effect of further improvement in the quality of life after program completion.

In this study, the PANAS scale scores indicated significant reductions in negative emotions up to 1 week after the program, in line with reports that forest exposure alleviates stress [5,6]. Positive mood improved significantly only immediately after program completion. However, there was no significant change in positive emotions on the PANAS

scale up to 1 week after completion. Similar findings were observed in previous studies, i.e., a more positive mood immediately after program completion. However, in another study [18], significant effects on positive emotions, as revealed by the PANAS scale, were observed immediately after the intervention and for up to 1 month after. Although we thus confirm that forest therapy and forest exposure can reduce negative emotions in the short term, we did not find a sustained effect.

Forest therapy lowered blood pressure and HRV in this study. However, all values were within the normal range before the program. Other studies found that forest therapy decreased blood pressure for up to 3 days, or 1, 4, or 8 weeks; forest phytoncides and negative ions reduced blood pressure, as did meditation and walking [19]. However, compared to controls, the effects were not significant. Two prior studies investigating health status before and after forest therapy also observed significant improvements in HRV [20,21] but no long-term effects, similar to our results. Future research should examine long-term changes in HRV.

Few quantitative studies have explored the impact of forest therapy on physical activity. However, qualitatively, it has been suggested that such therapy increases exercise motivation [22]. The duration of moderate-intensity exercise increased significantly 1, 2, and 4 weeks after the end of our program. Moderate-intensity exercise includes yoga and brisk walking during forest therapy; it can thus be assumed that the time spent on such exercises increased after the program.

Previous studies showed that walking or physical activity in a natural environment increased physical activity and social and psychological health [23]. Physical activity is associated with a sense of accomplishment, better self-perceived health, and stress relief. Physical activity in green spaces encourages positive behavior [24]. Those who initially enjoy forests consistently seek them out [25]. Forest therapy promotes long-term physical activity and enhances the forest experience.

## 5. Limitations

First, loss to follow-up was significant; 53 participants were lost after one week, 67 after two weeks, and 82 after four weeks. In this study, while the final remaining number of participants is 17 (after 4 weeks), previous research examining the effects of forest therapy programs suggests that studies have been conducted with even fewer participants than 17 and, in some cases, with as few as 10 participants, yielding significant health improvement effects [26,27]. This study initially targeted 99 participants, and while the number decreased during the long-term investigation, we believe that conducting a long-term follow-up is important to provide new and distinct results compared to previous research. The second limitation is the difficulty in determining whether the observed effects in this study originated from indoor or outdoor programs. This is because all participants had already chosen their preferred indoor and outdoor programs at the forest site before agreeing to participate in the study. However, the forest therapy program includes at least one indoor program and one or more outdoor programs over a 2-day, 1-night period. Therefore, since all participants experienced both indoor and outdoor programs at least once, it can be said that the effects of both indoor and outdoor programs were considered. Supplementary Table S1 shows the health parameters by group before and after the program. Another limitation was that the study was conducted during the COVID-19 pandemic; some subjects had emotionally demanding occupations, which may explain the minimal improvement in mood.

## 6. Conclusions

This study investigates the enduring impact of forest therapy on health at 1, 2, and 4 weeks following the conclusion of the forest therapy program, as well as its influence on changes in physical activity behavior. We confirmed that the indicators showing significant effects for 1 week after program completion are anxiety, mood, and quality of life. The indicators showing significant effects for 2 weeks are the negative mood indicators (PANAS)

and for 4 weeks are depression, METs, and moderate-intensity exercise time. It represents the first study to confirm that the effects of forest therapy on depression reduction and the increase in moderate-intensity exercise time can persist for up to 4 weeks. In future research, it is suggested that the findings from this study can serve as a valuable reference for tool selection during the study design phase, and further examination of changes in physical activity behavior is proposed in subsequent studies.

**Supplementary Materials:** The following supporting information can be downloaded at: https://www.mdpi.com/article/10.3390/f14112236/s1, Table S1: Changes in the health status of each group before and after the program; Table S2: Changes in the HRV status of 99 participants before and after the program; Table S3: Changes in the health status by sex, age, and BMI category group before and after the program.

**Author Contributions:** Conceptualization, H.-r.C.; methodology, H.-r.C.; validation, H.-r.C., S.-i.C., S.P. and G.K.; formal analysis, H.-r.C.; investigation, H.-r.C., I.C. and Y.Y.C.; writing—original draft preparation, H.-r.C.; writing—review and editing, H.-r.C., I.C. and Y.Y.C.; visualization, H.-r.C., I.C. and Y.Y.C.; supervision, S.-i.C.; project administration, S.-i.C. All authors have read and agreed to the published version of the manuscript.

**Funding:** This study was supported by the R&D Program for Forest Science Technology (project no.: 2021388A00-2123-0102) of the Korea Forest Service (Korea Forestry Promotion Institute).

**Data Availability Statement:** The published articles included in our analysis are publicly available.

**Conflicts of Interest:** The authors declare no conflict of interest.

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
