# Peer review of "Effects of a Forest Therapy Program on Physical Health, Mental Health, and Health Behaviors"

_forests, doi:10.3390/f14112236_

Round 1

Reviewer 1 Report

Comments and Suggestions for Authors

Summary:

Two aspects of the article are valuable, one is that it conducts a long-term study of forest therapy, and this type of tracking long-term health benefit study contributes to current research on forest therapy. The other focuses on the effects of forest therapy on physical activity, which plays an important role in health, and it is innovative to start with the perspective that forest therapy affects long-term exercise behaviour.

General comments:

Pg2 L66: The number of responders at follow-up is too small as a proportion of the total number of participants (especially after 2 weeks and after 4 weeks), concerns that it does not give a more accurate picture of the effects after forest therapy for the majority of participants, and that as many people as possible should be included in the follow-up.

Pg2 L70: Why is 2.2 titled "Systematic review"? Systematic review is not appropriate as a subheading for Materials and Methods. If the survey tools used in previous studies are to be described and evaluated in detail, they should be placed in the introduction. Otherwise, there is no need to split into "2.2.1. Survey tools and protocols" and "2.2.2. Survey tools". In addition, the review looks more like a narrative review than a systematic review.

Pg3 L101: The section "2.2.3. Data collection" does not describe the specific activities of the forest therapy. If this section had been combined with 2.1 Pilot Study, it would have been easier for the reader to get a full picture of the study. The fact that the participants performed different forest activities is a big problem for this study.

Pg3 L101: Physiological data were not continuous measurements taken over the course of the forest therapy.

Pg5 L198: HRV contains both time-domain indicators and frequency-domain indicators. time-domain indicators are usually reported as SDNN or RMSSD and frequency-domain indicators are usually reported as HF,LF or HF/LF in past studies. Please explain how the HRV sub-indicators in this paper are calculated and how they relate to the SDNN, RMSSD, HF, LF and HF/LF indicators reported in most studies.

Why there is no "Conclusion" section.

Reviewer 2 Report

Comments and Suggestions for Authors

Please find all the comments and suggestions in the attached file.

Reviewer 3 Report

Comments and Suggestions for Authors

I am very impressed by many facets of this paper.  While many studies have examined the effects of forest therapy on aspects of mental health, few have studied the effects of such activities simultaneously in mental and physical health, and health behavior.  The authors were conscientious in their selection of evaluation tools based on their use in previous studies.  The research protocol is sound and I appreciate that it included follow-up surveys one, two, and four weeks after the intervention.  In addition, the data analysis is thorough and is presented in a clear format in the tables.

There are, however, two glaring problems with this study, both of which are mentioned in the Limitations section (by the way, this section is not titled and needs to be).  The first problem is the dropout rate among participants post intervention: 53 participants were lost after 1 week, 67 after 2 weeks, and 82 after 4 weeks.  This left very few data points for the latter surveys which calls their validity into question.  Secondly, “not all participants followed the same programs.”  It is inappropriate to lump together those participants who engaged in forest walking with those who conducted meditation exercises.  If possible, I would suggest that the data be reanalyzed with the results for participants in these two groups considered separately.

Round 2

Reviewer 3 Report

Comments and Suggestions for Authors

While the authors have made considerable efforts to address issues in the Introduction, Materials and Methods, and Results, I feel that the fundamental flaws in the research that I highlighted in my previous review cannot be altered: i.e., that the majority of participants dropped out of the study in the weeks after completing their activities, and that the researchers did not distinguish between study participants who meditated and those who engaged in forest therapy.   While both of these shortcomings are pointed out in the Limitations section (which still needs a title), the value of publishing this study remains an open question.

The impact of the limitations on drawing conclusions be stated more overtly, and that a thorough copy editing be conducted to address grammatical flaws.

Comments on the Quality of English Language

This paper would be improved by a thorough copy editing to eliminate grammatical errors.
